# Context Compression for Auto-regressive Transformers with Sentinel Tokens

**Siyu Ren**      **Qi Jia**
Shanghai Jiao Tong University, China
{roy0702, Jia_qi}@sjtu.edu.cn

**Kenny Q. Zhu**[*]
University of Texas at Arlington, USA
kenny.zhu@uta.edu

## Abstract

The quadratic complexity of the attention module makes it gradually become the bulk of compute in Transformer-based LLMs during generation. Moreover, the excessive key-value cache that arises when dealing with long inputs also brings severe issues on memory footprint and inference latency. In this work, we propose a plug-and-play approach that is able to incrementally compress the intermediate activation of a specified span of tokens into compact ones, thereby reducing both memory and computational cost when processing subsequent context. Experiments on both in-domain language modeling and zero-shot open-ended document generation demonstrate the advantage of our approach over sparse attention baselines in terms of fluency, n-gram matching, and semantic similarity. At last, we comprehensively profile the benefit of context compression on improving the system throughout. Code is available at https://github.com/DRSY/KV_Compression.

## 1 Introduction

The Transformer architecture (Vaswani et al., 2017) has become the underpinning component of modern large language models (LLMs) (Radford et al., 2019; Devlin et al., 2019; Zhang et al., 2022; Touvron et al., 2023) in recent years. However, the quadratic computational complexity and memory footprint of the attention mechanism have largely limited applying Transformers to increasingly long contexts with constrained computing resources.

To mitigate such issues, prior works (Child et al., 2019; Beltagy et al., 2020; Zaheer et al., 2020; Kitaev et al., 2020) have explored an assortment of efficient Transformer variants, mostly by replacing the original quadratic attention operation with various forms of linearized approximation. Though promising, large-scale pre-training and specialized

---

[*] The corresponding author.

CUDA kernels are typically required for these models to achieve performance comparable to off-the-shelf LLMs and fulfill real efficiency gains.

In this work, we aim to improve the efficiency of existing Transformer-based LLMs without any architectural changes. Specifically, we focus on the key-value cache, which accounts for the majority of memory footprint and data movement (I/O) cost when dealing with increasingly long input using LLMs. We propose a plug-and-play approach to incrementally compress the key-value memory of a contiguous span of tokens into compact ones. Specifically, we introduce a pair of special sentinel tokens *<CL>* and *<CR>* into the vocabulary of LLMs and use them to mark the boundary of the span to be compressed. During training, we modify the causal attention mask such that future tokens after *<CR>* are precluded from attending to tokens between *<CL>* and *<CR>*. By continually training LLMs with the next token prediction objective, the model learns to extract and condense task-relevant information of the bounded span into the ending sentinel token. The reduced context length alleviates both memory and computational costs when processing subsequent tokens, thereby improving system throughput with larger batch sizes and faster decoding speed during inference.

We conduct experiments on the WikiText-2 language modeling benchmark and show that our approach is generalizable to LLMs with various sizes (from 1.3B to 3B) and position encoding schemes such as absolute position embedding (Devlin et al., 2019) and rotary position encoding (Su et al., 2021). Compared to sparse attention baselines, our approach is able to effectively *compress historical key-value cache* with significantly reduced degradation in perplexity. Moreover, we demonstrate that our approach outperforms sparse attention in *zero-shot open-ended document generation* across different compression ratios as evaluated by perplexity, ROUGE (Lin, 2004), and

BERTScore (Zhang et al., 2019). Finally, we empirically demonstrate that context compression is able to confer considerable improvement in *system throughput* for text generation.

## 2 Background and Related Work

In this section, we present necessary background knowledge as well as drawing distinctions of our approach to existing literatures.

**Complexity of Attention Mechanism** The self-attention mechanism of LLMs produces a query vector for each token to select and retrieve information from all previously computed key and value vectors. Thus, a key-value memory is required at runtime to store past information. Given a pretrained LLM with $M$ Transformer layers, $H$ attention heads, per-head hidden dimension of $d_{head}$, batch size $b$, and current context length $L$, the model stores key and value vectors computed so far as a tensor of shape $(M, 2, b, H, L, d_{head})$. During auto-regressive decoding, the size of key-value cache grows linearly with the context length $L$, leading to significant increase in memory footprint and latency.

**Efficient Transformers** Various efficient variants of Transformers have been proposed to address the problematic complexity of attention. Sparse Transformer (Child et al., 2019) limits the receptive field of each token to a local window. Longformer (Beltagy et al., 2020) and BigBird (Zaheer et al., 2020) introduce additional randomly and globally accessible tokens to compensate for the information loss. Linear Transformer (Katharopoulos et al., 2020) reformulates the self-attention as a linear dot-product of kernel feature maps and make use of the associativity property of matrix products to reduce the complexity from $O(N^2)$ to $O(N)$. Nevertheless, these approximations were shown to degrade the expressiveness of full-attention and have been upstaged by GPT-like LLMs recently. In this work, however, we tackle this problem by compressing lengthy input into more compact representations, which is orthogonal to on-going efforts to architectural optimization.

**Gisting Tokens** Mu et al. (2023) explored using gisting tokens to compress textual prompts during instruction tuning. They showed that verbose task instructions can be compressed into much shorter ones. However, their approach is only applicable to prefix text of around 20 tokens. In contrast, we aim

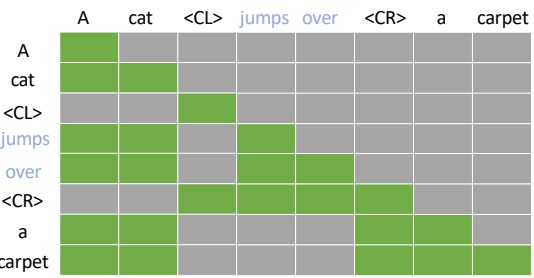

Figure 1: Modified attention mask for context compression. Green and grey boxes indicate positions that are allowed and disallowed to be attended to (attention direction from row → column).

for a compression scheme with more flexible compression choices and larger compression ratios.

## 3 Approach

In this section, we present a plug-and-play approach to compress the full-length contextual representations into shorter ones while preserving information necessary to accomplish the end task.

### 3.1 Context Compression with Sentinel Tokens

To alleviate the computational and memory intensity of key-value memory when processing increasingly long context, we introduce two sentinel tokens *<CL>* and *<CR>* into the vocabulary of LLM, that are placed around a contiguous span of tokens of which the corresponding key-value memory will be compressed.

In order to consolidate information of tokens surrounded by *<CL>* and *<CR>*, we design a modified causal attention mask to facilitate such behaviour. An illustrative example is shown in Figure 1. Specifically, *<CL>* serves a start-of-compression symbol and can only attend to itself. Normal tokens (tokens except for *<CL>* and *<CR>*) have access to all previous *<CR>* and normal tokens. This ensures that the contextual representations are built upon both compressed (lossy) and complete (lossless) past information. *<CR>* then act as a form of information selection and retrieval from contextual representations of surrounded tokens. This modified masking scheme, combined with task-specific fine-tuning of LLM, encourages distilling task-relevant information of potentially long token sequences into a compact representation, thereby reducing the size of key-value memory required for subsequent processing.

| Model | Method | Compression Ratio ($r$) | | | | | | | | | |
|---|---|---|---|---|---|---|---|---|---|---|---|
| | | 0.0 | 0.1 | 0.2 | 0.3 | 0.4 | 0.5 | 0.6 | 0.7 | 0.8 | 0.9 |
| OPT-1.3B | Scattered Attention | 15.0 | 18.2 | 22.6 | 28.2 | 35.8 | 47.2 | 59.6 | 81.0 | 106.4 | 151.6 |
| | Local Attention | 15.0 | 15.1 | **15.2** | 15.7 | 16.3 | **17.2** | 18.2 | 19.9 | 23.0 | 29.8 |
| | KV Compression | 15.0 | **15.0** | 15.2 | **15.6** | **15.9** | 17.3 | **17.8** | **18.0** | **18.1** | **18.3** |
| OPT-2.7B | Scattered Attention | 13.1 | 16.2 | 19.8 | 25.5 | 32.4 | 41.7 | 56.3 | 75.7 | 101.4 | 142.3 |
| | Local Attention | 13.1 | **13.1** | **13.4** | 13.9 | 14.4 | 15.3 | 16.1 | 17.5 | 20.1 | 26.6 |
| | KV Compression | 13.1 | 13.2 | 13.5 | **13.7** | **14.0** | 15.3 | **15.8** | **16.0** | **16.1** | **16.3** |
| RedPajama-3B | Scattered Attention | 11.2 | 13.1 | 16.5 | 20.7 | 26.6 | 34.9 | 47.2 | 65.7 | 91.6 | 135.9 |
| | Local Attention | 11.2 | **11.3** | 11.5 | **11.8** | 12.2 | 12.9 | 13.6 | 14.8 | 17.2 | 22.9 |
| | KV Compression | 11.2 | 11.4 | **11.5** | **11.8** | **11.9** | **12.3** | **13.5** | **13.7** | **13.8** | **14.0** |

Table 1: Perplexity (the lower the better) of three LLMs on WikiText-2 language modeling benchmark.

**Input Transformation**   Given an input $x$ with $L$ tokens, we define compression ratio $r$ as the percentage of the tokens to be enclosed by one or more pairs of *<CL>* and *<CR>*, and max compression length $l$ as the largest number of tokens to be compressed by a single pair of *<CL>* and *<CR>* tokens. We repeatedly sample span of tokens with length conforming to a uniform distribution $U(2, l)$ as compression target, until ratio $r$ is reached. But these spans must not overlap. We tie the position encoding of *<CL>* and *<CR>* to their previous token, such that they don't occupy valuable positions for LLMs using absolute position embedding.

**Training**   We opt for a light-weight procedure for adapting LLMs to downstream tasks through fine-tuning. Specifically, all parameters of the LLM are frozen except for the embeddings of *<CL>* and *<CR>* and LoRA (Hu et al., 2021) modules applied to all attention layers. This not only eases the training process of models with billions of parameters but also ensures the inherent language modeling ability and parametric knowledge in feedforward networks are intact (Geva et al., 2021).

**Manipulating Key-Value Memory for Inference-time Compression**   Given a text piece as prefix, we apply the same input transformation strategy defined above to reach a specified compression ratio. To realize perceivable memory and computation reduction, the key-value cache of tokens enclosed by sentinel tokens is freed from GPU in a progressive manner (e.g., using a for-loop over blocks of tokens) if the original prefix is too long to fit into GPU. Otherwise we just feed the transformed prefix through the model and free the key-value cache of marked tokens in one go.

| Method | Compression Ratio ($r$) | | |
|---|---|---|---|
| | 0.7 | 0.8 | 0.9 |
| Local Attention | 23.1 | 25.2 | 29.5 |
| KV Compression | **21.8** | **22.0** | **22.5** |

Table 2: Perplexity (the lower the better) of RedPajama-3B on WritingPrompts test set.

## 4   Experiments

We mainly evaluate our approach on language modeling task and then explore its zero-shot generalization ability on open-ended document generation. Finally, we quantitatively measure the effect of context compression on system throughput.

### 4.1   Language Modeling

**Benchmark**   We mainly conduct experiments on the Wikitext-2 (Merity et al., 2016) dataset consisting of Wikipedia articles, which has been widely used for the evaluation of language modeling. We report token-level (excluding *<CL>* and *<CR>* for KV Compression) average perplexity on the test set as the evaluation metric. We also report results on the WritingPrompts (Fan et al., 2018) dataset to investigate the generality of our method.

**Models**   To demonstrate the generality of our approach, dubbed KV Compression, we adopt OPT-1.3B, OPT-2.7B (Zhang et al., 2022), and RedPajama-3B (Computer, 2023) as three LLMs with different sizes and position encoding methods.

**Baselines**   We compare our approach with two sparse attention baselines: (1) Scattered Attention which samples positions that permits attention from a Bernoulli distribution $B(1\text{-}r)$; and (2) Local Attention (Beltagy et al., 2020) which restricts the attention of token $x_t$ to be within $(\lfloor t * r \rfloor, t)$. For KV Compression, we set $l$ as 25 throughout the ex-

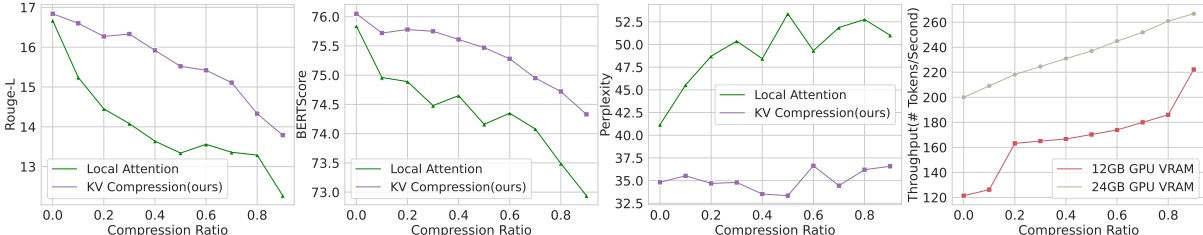

Figure 2: RedPajama-3B on open-ended generation on 200 sampled C4 documents. Generation quality is measured by fluency (perplexity), n-gram matching (ROUGE-L), and semantic similarity (BERTScore). We report system throughput as the number of tokens generated per second with different maximum GPU VRAM.

periments. Our proposed approach, alongside the sparse attention baselines we have selected, shares a common goal: enhancing the ratio of computation to memory access by curtailing the stored key-value cache. For detailed training setup please refer to Appendix A.

**Results**   The results on WikiText-2 are shown in Table 1. As the compression ratio $r$ gets larger, all three methods result in an increase in perplexity either due to: (1) important tokens are out-of-scope for attention, or (2) the capacity of single *<CR>* token is insufficient to encapsulate the full information of compressed token span. We observe that Local Attention substantially outperforms Scattered Attention, suggesting the importance of local context for language modeling. KV Compression achieves the best perplexity across various compression ratios. Notably, at high compression ratios, e.g., $r \geq 0.7$, KV Compression incurs significantly fewer degradation in perplexity compared to Local Attention, demonstrating its advantage in compressing scattered local information meanwhile keeping coherent global information. In Table 2, we report the perplexity of RedPajama-3B on WritingPrompts dataset using Local Attention and the proposed KV Compression, with compression ratio from {0.7, 0.8, 0.9}. KV Compression achieves consistently lower perplexity compared to Local Attention, indicating its superiority as a domain-generalizable method for context compression.

### 4.2   Zero-shot Open-ended Generation

**Data**   We randomly select 200 documents from C4 (Raffel et al., 2020) validation set for evaluation. We use the leading 128 words as prefixes and treat the next 64 words as ground-truth references.

**Models**   We directly take the RedPajama-3B model trained on Wikitext-2 to perform zero-shot

open-ended document generation. Given a prefix text $p$, nucleus sampling (Holtzman et al., 2019) is used to generate a completion $c$ for it.

**Baselines**   Because Scattered Attention performs poorly according to Table 1, we only compare KV Compression with Local Attention with compression ratio $r$ ranging from 0.0 to 0.9 applied to the prefix $p$ using input transformation defined in Section 3 for KV Compression and restricted attention defined in Section 4. For Local Attention to achieve inference-time compression, it amounts to maintaining a FIFO queue to store the key-value cache: as the time step during generation increases, old key-value memory in the queue is popped out and newly generated key-value memory is pushed in.

**Evaluation Metrics**   We evaluate the quality of generated completions from three aspects: (1) fluency, (2) n-gram matching with ground-truth continuation, and (3) semantic similarity with ground-truth continuation. Fluency is evaluated by perplexity computed from Pythia-1B (Biderman et al., 2023) model pre-trained using C4. N-gram matching and semantic similarity are measured by ROUGE-L and BERTScore (Zhang et al., 2019) respectively. To account for the randomness induced by nucleus sampling, for each prefix, we generate 8 completions and report the average results.

**Results**   Figure 2 shows that, as $r$ increases, the generated completions of Local Attention tend to diverge from the original topic, leading to decreased ROUGE-L/BERTScore and increased perplexity. Again, KV Compression excel at preserving relevant information and can still generate decent-quality continuations up to 0.5 compression ratio. Notably, KV Compression can generate fluent text even when the prefix is extremely compressed. The reason is that, during training, Local Attention receives rapidly faded information

from the distant past, making the discourse structure for subsequent generations incoherent. On the contrary, KV Compression better preserve such information by consolidating it into sentinel tokens.

### 4.3 Throughput Gain from Context Compression

With KV Compression, the key-value cache corresponding to tokens enclosed by sentinel tokens can be freed from memory. In this way, it permits a larger batch size and improves the system throughput in return. To quantitatively measure the impact of context compression on throughput, we conduct experiments on open-ended generation by testing with different maximum available GPU VRAM. The full length of the dummy input prefix is set to 800 and we use RedPajama-3B with nucleus sampling to generate a continuation for it. The throughput of the text generation system is defined as the number of tokens generated per second. The results are shown in the bottom right of Figure 2. We can see that, at extreme compression ratios (e.g., $r \geq 0.8$), context compression offers more than 1.5x throughout improvement with 12GB GPU VRAM and a slightly smaller 1.3x improvement with 24GB GPU VRAM. At moderate compression ratio (e.g., $r \approx 0.5$), KV compression is still able to deliver 1.2x-1.4x throughout improvement while only suffering from mild quality drop (Section 4.2). More visualized memory-compression ratio correlation is deferred to Appendix D.

## 5 Conclusion

In this work, we propose a simple yet effective approach that enables LLMs to summarize the key-value memory of specified span of tokens. Experiments on language modeling and open-ended generation demonstrate that our approach substantially outperforms sparse attention baselines in terms of information compression and generation quality.

## Limitations

In the current evaluation setup, we apply the same strategy used for training to select spans of tokens to be enclosed by *<CL>* and *<CR>* for inference-time context compression. However, different text pieces may display different degrees of importance for downstream tasks. For instance, a grammatical and semantic complete noun phrase can be more compressible than an ungrammatical one that contains only partial linguistic units. Though our input transformation procedure theoretically includes text spans of all possible linguistic structures, it may still benefit from an elaborately designed strategy/algorithm for selecting compression targets in a given context.

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

## A Implementation Details

**Language Modeling** For language modeling experiments, we train OPT-1.3B, OPT-2.7B, and RedPajama-3B with the next token prediction objective. The loss is computed on all tokens except for newly added *<CL>* and *<CR>* tokens. For tokens right before *<CL>* and *<CR>*, we adjust their ground-truth label to be tokens right after the corresponding *<CL>* and *<CR>* tokens.

The trainable parameters for all LLMs include two token embeddings of *<CL>* and *<CR>*, and LoRA modules applied to all attention layers. The rank of weight matrices of LoRA is set to 16. The percentage of trainable parameters takes about 5% of the total parameters of the LLM. We use AdamW (Loshchilov and Hutter, 2017) optimizer with a 2e-5 learning rate. The batch size is set to 12 and the maximum sequence length during training is set to 256 due to the limited computation budget. The implementation is based on Huggingface transformers (Wolf et al., 2020) library and all experiments are conducted using a single RTX 3090 GPU.

**Open-ended Document Generation** For open-ended document generation, we use nucleus sampling with top p=0.9. Due to the randomness induced by sampling, we perform 8 times nucleus sampling for each compressed prefix and report the average evaluation metrics adopted in the main paper. This helps reduce variance and ensures a more reliable evaluation result.

## B Generalization Ability of KV Compression

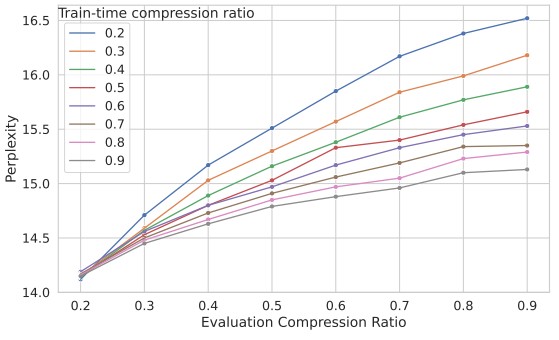

Figure 3: Generalization performance of different train-time compression ratios.

In our proposed KV Compression, an LLM is trained on transformed data in which the percent of tokens surrounded by sentinel tokens takes a specified ratio of the total number of tokens. Here we study the generalization performance of different train-time compression ratios and explore the best practice. We visualize the perplexity of RedPajama-3B trained with different compression ratios at varying test-time compression ratios.

Figure 3 illustrates the results. We can see that training with a higher compression ratio always results in better generalization ability across various test-time compression ratios. The reason is that, at a small compression ratio, language models can sometimes exploit limited local context and can still reliably predict the next token. In this case, the sentinel tokens are not guaranteed to acquire the desired context compression ability. When the majority of context is not allowed to be attended to, the sentinel tokens are forced to cultivate enough compression ability in order to minimize the loss function.

## C Memory Usage with Varied Compression Ratio

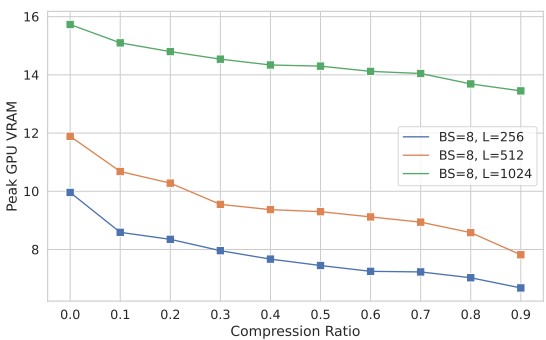

Figure 4: Peak cached memory profiled using Pytorch when using RedPajama-3B to produce a 100-token continuation for variable-length prefixes.

We have shown that context compression confers decent improvement in system throughput, especially for moderate-sized GPUs. Here we report detailed memory usage assuming a practical scenario similar to multi-turn dialogue: the prefix length (length of historical dialogue) gradually increases, and the model is asked to output a response of roughly 100 tokens. To maximize the throughput of dialogue service, we assume the model is simultaneously generating responses for multiple instances, i.e., a batch size larger than one.

The visualized memory usage is shown in Figure 4. Context compression is able to reduce more than

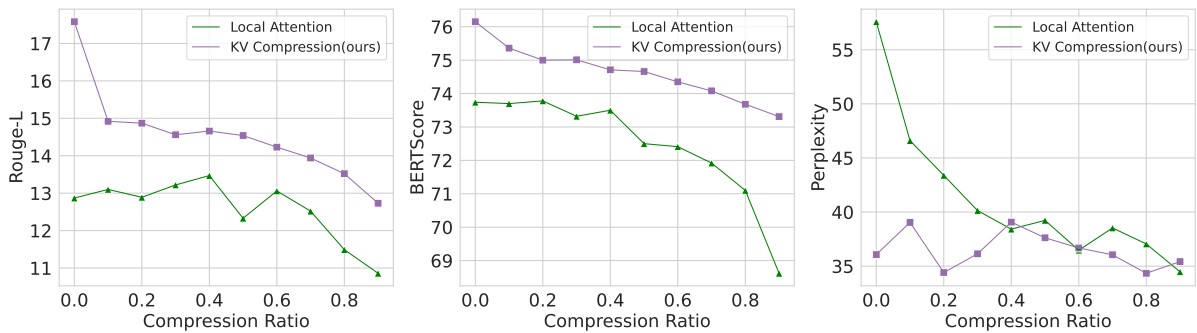

Figure 5: OPT-1B on open-ended generation on 200 sampled C4 documents. Generation quality is measured by fluency (perplexity), n-gram matching (ROUGE-L), and semantic similarity (BERTScore).

3GB of peak cached GPU memory. In practice, this translate to the feasibility of generating more tokens for a single instance and enabling more instances in parallel.

## D   Open-ended Generation Results of OPT

Figure 5 summarizes the results of OPT-1B on open-ended document generation. Compared to that of Redpajama-3B (Figure 2), the generation quality of OPT-1B is substantially lower. Comparing different context compression methods, KV Compression performs uniformly better across all three evaluation metrics.