# OpenReview forum: "Context Compression for Auto-regressive Transformers with Sentinel Tokens"
_EMNLP/2023/Conference — EMNLP 2023 Main_

### Official Review · Reviewer_BNcU · 2023-08-04

**Soundness:** 3

**Excitement:**

3: Ambivalent: It has merits (e.g., it reports state-of-the-art results, the idea is nice), but there are key weaknesses (e.g., it describes incremental work), and it can significantly benefit from another round of revision. However, I won't object to accepting it if my co-reviewers champion it.

**Justification For Ethical Concerns:**

Justification for Ethical Concerns

**Paper Topic And Main Contributions:**

In this paper, the authors proposed a key-value context compression approach that places two sentinel tokens around a contiguous span of tokens and defines the position that are allowed and disallowed to be attended to using causal attention masking.

**Questions For The Authors:**

- The caption in figure 1 does not clearly indicate the color encodings of the attention mask.
- The proposed method is compared against two methods only, any reason for not considering more methods?
- The proposed method is only experimented on one dataset, Wikitext-2. Do you expect a different result if you conduct your experiment on prompts dataset, like chatgpt-prompts? Do you think your result will be comparable with the work of “Learning to Compress Prompts with Gist tokens” by Jesse Mu, Xiang Lisa Li, Noah Goodman.

**Reasons To Accept:**

The authors proposed a novel key-value context compression method and conducted comparisons against one dataset, two methods and three models. The proposed approach shows improvement compared with other sparse compression methods.


**Reasons To Reject:**

Two major reasons to reject this paper.
- The proposed approaches are combinations of well-known algorithms and the metrics are already popular in the community. The Author's contributions are marginal at best.
- No theory behind. Formal definition and theoretical background of the proposed method are missing in the paper.


**Reproducibility:**

3: Could reproduce the results with some difficulty. The settings of parameters are underspecified or subjectively determined; the training/evaluation data are not widely available.

**Reviewer Confidence:**

4: Quite sure. I tried to check the important points carefully. It's unlikely, though conceivable, that I missed something that should affect my ratings.

**Typos Grammar Style And Presentation Improvements:**

- Typos, Abstract section line 013  “flucency" --> "fluency"
- Typos, Abstraction section line 016 “thoughout” → “throughput”

---

> ### Author Rebuttal · Authors · 2023-08-27
>
> We appreciate the valuable feedback provided by the reviewer. Our responses to Reviewer's questions are listed below:
>
> **Q**: "The caption in Figure 1 does not clearly indicate the color encodings of the attention mask."
> **A**: We apologize for the confusion and will fix it in the revision. In Figure 1, the color in $i$-th row indicates whether one token can be attended by the $i$-th token or not. Green-colored tokens are allowed to be attended to while grey-colored tokens are disallowed. For example, for the token 'carpet', there are five tokens that it can attend to: 'A', 'cat', '<CR>', 'a', and 'carpet'.
>
> **Q**: "The proposed method is compared against two methods only, any reason for not considering more methods?"
> **A**: The end goal of our proposed method is to curtail the length of the key-value cache so that both memory access cost and memory footprint are ameliorated. Therefore, our method is directly comparable to sparse attention-based methods that restrict the receptive field of each token to a smaller subset of tokens, thus reducing the size of the key-value cache required to be stored. We didn't compare against methods that reduce key-value cache from orthogonal perspectives, e.g., decrease the number of attention heads.
>
>
> **Q**: "The proposed method is only experimented on one dataset, Wikitext-2. Do you expect a different result if you conduct your experiment on prompts dataset, like chatgpt-prompts? Do you think your result will be comparable with the work of “Learning to Compress Prompts with Gist tokens” by Jesse Mu, Xiang Lisa Li, Noah Goodman."
> **A**: In addition to using WikiText-2 for language modeling experiments, we also used C4 to evaluate the effectiveness of different context compression methods on open-ended generation using C4. Language modeling dataset like WikiText contains data with presumably more inherent variations than prompts dataset. Thus, we expect our method to work comparably well on domains beyond the general language modeling corpus. To validate this point, we report here additional results(ppl) on WritingPrompts dataset using Redpajama-3B under compression ratios of 0.7, 0.8, and 0.9:
>
> | Method/Compression Ratio |  0.7 |  0.8 |  0.9 |
> |:------------------------:|:----:|:----:|:----:|
> |      Local Attention     | 23.1 | 25.2 | 29.5 |
> |      KV Compression      | 21.8 | 22.0 | 22.5 |
>
> The gist token method proposed by Mu et at. is restricted to compressing a moderate-length prefix, e.g., prompt/instructions for supervised instruction tuning. When the target compression ratio is high, such prefix compression methods can hardly retain information needed for subsequent processing. In contrast, our method is more flexible and can retain much coherent information flow even under high compression ratio.

---

### Official Review · Reviewer_UZc8 · 2023-08-04

**Typos Grammar Style And Presentation Improvements:** L159 => span**s** of tokens
**Soundness:** 3

**Excitement:**

3: Ambivalent: It has merits (e.g., it reports state-of-the-art results, the idea is nice), but there are key weaknesses (e.g., it describes incremental work), and it can significantly benefit from another round of revision. However, I won't object to accepting it if my co-reviewers champion it.

**Paper Topic And Main Contributions:**

This paper introduces a novel method to compress self-attention maps during LLM-based text generation. The method consists in using two sentinel tokens that are randomly placed along the sequence around spans that will then be considered as single entities in the self-attention operations. Experiments show that after a short and efficient finetuning of the LLMs adapted with the sentinel mechanism, models are able to maintain satisfying generation capabilities across compression ratios, which is not the case for compared compression approaches. The authors finally expose the throughput increase allowed by their method, hence demonstrating the relevance of their method.

**Questions For The Authors:**

1 - Do you think it would be easy to write an efficient CUDA kernel for your specific method?

2 - Did you run experiments to verify how the randomness of the selection process impacts the variability of the performance?

3 - Is a given compression ratio consistently implying a given throughput across methods and implementations? In other words, do you think that for a given type of implementation (optimized CUDA kernels, fast attention, ...) a given compression ratio can yield a similar GPU throughput between Local Attention and KV compression? If there is such a relation, I think it would be interesting to mention it in your paper. If not, I think it would be interesting to compare *achievable throughput* and generation performance.

**Reasons To Accept:**

This paper presents a novel method, and displays extensive experiments that convincingly demonstrate its relevance for open-ended text generation. The core idea is elegant and paves the way for future work. It could be impactful for the community as it allows more time and memory-efficient generation using LLMs at a low engineering and performance cost. The writing of the paper is compact and straightforward, which is satisfying.

**Reasons To Reject:**

This paper is missing an important discussion in my opinion, which is the relation between compression ratio and throughput *across different methods and implementations of these methods*. I am not sure whether it would affect final conclusions positively or negatively, but I think it is a crucial point to demonstrate the actual usefulness of the approach. For instance, it is clear that local attention throughput can be increased by specific CUDA kernels, and thus I feel that it would be interesting to compare *Generation capabilities vs. Throughput* across methods in the same fashion as in Figure 2. After rebuttal, the authors addressed this concern for the inference part, but I believe the clarity of the paper could be improved on that matter.

Another potential drawback - that is addressed in the limitations section - is the randomness of the span selection process. It seems to me that it might affect the robustness of the generation performance. It would have been interesting to try to answer the following question: "How often the randomness of span selection hurts generation capabilities?" by studying worst-case scenarios, and variability of the performance across seeds/samples.

Overall, while this paper **is relevant as is**, it would benefit from more thorough analysis and I think a long paper may be more suited for this work.

**Reproducibility:**

5: Could easily reproduce the results.

**Reviewer Confidence:**

3: Pretty sure, but there's a chance I missed something. Although I have a good feel for this area in general, I did not carefully check the paper's details, e.g., the math, experimental design, or novelty.

---

> ### Author Rebuttal · Authors · 2023-08-27
>
> We appreciate the valuable feedback provided by the reviewer. Our responses to Reviewer's questions are listed below:
>
> **Q**: "relation between compression ratio and throughput"
> **A**: Among methods that reduce the length of key-value cache, the notable contribution of our proposed KV Compression is that it offers the flexibility to compress selected spans of tokens(in contrast to all past information beyond the window size in the Local Attention method), therefore resulting in better language modeling ability(reflected by PPL) and generation quality(reflected by PPL, ROUGE, and BERTScore). In Section 4.3, we profile the gain of throughput(the number of tokens generated per second) by compressing the key-value cache of ***prefix*** tokens, which essentially decreases the Memory Access Cost(MAC) of repeatedly reading from/writing to HBM. Specifically, we use different methods to encode the prefixing 800 tokens and only keep (1-compression_ratio) of the key-value cache. Therefore, the change of reported throughput only reflects the impact of reduced MAC and is the same of different methods under the same compression ratio.
> The specific CUDA kernels for efficient attention implementation contribute to the inference efficiency ***from another aspect***. our proposed KV Compression can be readily combined with existing implementation, e.g., xformers' memory_efficient_attention, which we observe empirically offers more than 2x speed up than vanilla attention implementation.
>
> **Q**: "Do you think it would be easy to write an efficient CUDA kernel for your specific method?"
> **A**:  One can implement a specific CUDA kernel for our proposed method based on the existing Flash Attention kernel. The modification involves the outer-loop logic, i.e., for token spans enclosed by sentinel tokens, subsequent tokens except the ending sentinel token can be skipped.
>
> **Q**: "Did you run experiments to verify how the randomness of the selection process impacts the variability of the performance?"
> **A**:  Yes. To examine how the randomness in the selection process influences the generation quality, we ran our experiments on open-ended generation five times with different random seeds. The standard deviations for ppl, Rouge-L, and BERTScore across different compression ratios are 1.3, 0.6, and 0.5 respectively.

---

### Official Review · Reviewer_dqav · 2023-08-10

**Soundness:** 4

**Excitement:**

4: Strong: This paper deepens the understanding of some phenomenon or lowers the barriers to an existing research direction.

**Paper Topic And Main Contributions:**

This paper introduces a method for reducing memory and computational costs by compressing intermediate activations in auto-regressive Transformers using sentinel tokens. The proposed approach is shown to outperform sparse attention baselines in terms of fluency, n-gram matching, and semantic similarity through experiments on various tasks. Additionally, the paper provides a comprehensive analysis of the positive impact of context compression on system throughput.

**Questions For The Authors:**

A. Rationale for model selection?

B. Why is the code not open source?

**Reasons To Accept:**

The motivation is interesting.

The performance is significant.

**Reasons To Reject:**

Insufficient model selection

**Reproducibility:**

2: Would be hard pressed to reproduce the results. The contribution depends on data that are simply not available outside the author's institution or consortium; not enough details are provided.

**Reviewer Confidence:**

1: Not my area, or paper was hard for me to understand. My evaluation is just an educated guess.

---

> ### Author Rebuttal · Authors · 2023-08-27
>
> We appreciate the valuable feedback provided by the reviewers. Our responses to Reviewer's questions are listed below:
>
> **Q**: "Rationale for model selection?"
> **A**: Our primary deliberation revolves around encompassing various model scales, ranging from 1.3 billion to 2.7 billion parameters. Additionally, we take into account diverse position encoding schemes: absolute position encoding, as featured in OPT, and rotary position encoding, as employed in RedPajama, to demonstrate the generality of our approach. Extending to other LLMs is straightforward.
>
> **Q**: "Why is code not open source?"
> **A**: We have uploaded the source code in the supplementary material, including the main Python files and Bash scripts. We will open-source it after the paper is accepted.

---

### Official Review · Reviewer_xYfS · 2023-08-14

**Typos Grammar Style And Presentation Improvements:** 1. Line 078
**Soundness:** 3

**Excitement:**

3: Ambivalent: It has merits (e.g., it reports state-of-the-art results, the idea is nice), but there are key weaknesses (e.g., it describes incremental work), and it can significantly benefit from another round of revision. However, I won't object to accepting it if my co-reviewers champion it.

**Missing References:**

1. [Fast Transformer Decoding: One Write-Head is All You Need](https://arxiv.org/abs/1911.02150) presented a different attention mechanism that can compress KV cache by 2 order of magnitude
2. [FlashAttention: Fast and Memory-Efficient Exact Attention with IO-Awareness](https://arxiv.org/abs/2205.14135) presented a different computation algorithm to compute attention that saves memory and can be a drop-in replacement to the widely used attention algorithm.

**Paper Topic And Main Contributions:**

The paper presents a novel method to compress the memory consumption of KV cache and reduce computation time of attention in decoder Transformer models through the introduction of 2 sentinel tokens and fine-tune the model to compress the information between the span of 2 tokens to a single token allowing us to save only 1 token instead of the entire span of tokens. The authors show that using this method we can reduce the KV cache significantly with moderate accuracy loss, use less memory and increase throughput of the model.

**Questions For The Authors:**

A. Why were these baseline chosen? Aren't there baselines to compare to from previous papers?\
B. Not clear how the context compression is done at inference time? Are random spans of tokens are chosen?\
C. Same question as B regarding training.\
D. Can this method play well with Flash Attention to improve inference even further? If no, how your method compare with flash attention?

**Reasons To Accept:**

1. Easy to follow, mostly well explained and written paper.
2. Introduce a novel method to reduce memory consumption and increase throughput of existing pre-trained LLMs with a simple step of fine-tuning
3. Present empirical data to support the claims validity of the presented method.
4. Show actual benefit with current hardware and not only theoretical benefits.

**Reasons To Reject:**

1. Unclear baseline choice make it hard to compare this work to previous works, authors weren't clear why was those baselines chosen
2. Limited experimentation, the method was tested on a single benchmark and another generation task which (to my knowledge) is not widely used. Furthermore, relatively small and unpopular models were used for the evaluation making it hard to assume if this method can scale beyond the narrow experimentation presented.
3. No mention of open-sourcing implementation.

**Reproducibility:**

2: Would be hard pressed to reproduce the results. The contribution depends on data that are simply not available outside the author's institution or consortium; not enough details are provided.

**Reviewer Confidence:**

3: Pretty sure, but there's a chance I missed something. Although I have a good feel for this area in general, I did not carefully check the paper's details, e.g., the math, experimental design, or novelty.

---

> ### Author Rebuttal · Authors · 2023-08-27
>
> We appreciate the valuable feedback provided by the reviewers. Our responses to Reviewer's questions are listed below:
>
> **Q**: "A. Why were these baseline chosen? Aren't there baselines to compare to from previous papers?"
> **A**: The inference process of LLMs is inherently memory-bound. In the context of auto-regressive decoding, the predominant latency bottleneck is not attributed to actual computations, but rather to the repetitive movement of an increasingly expansive key-value cache. This cache is transferred from the High Bandwidth Memory (HBM) to the on-chip computation unit. Our proposed approach, alongside the sparse attention baselines we have selected, shares a common goal: enhancing the ratio of computation to memory access by curtailing the stored key-value cache.
>
> While Multi-query attention primarily concentrates on diminishing the count of attention heads, an endeavor that complements our approach, FlashAttention, introduces IO-aware modifications to the query-key computation scheme. Notably, both these methods run parallel to our aim of compressing the key-value pairs. In a subsequent version of our work, due credit will be given to these two aforementioned papers through proper citations.
>
> **Q**: "Not clear how the context compression is done at inference time? Are random spans of tokens are chosen?"
> **A**: During training, as stated in Section 3.1, we employ a random strategy to select spans of tokens to be compressed. We acknowledge that though more systematic span selection strategy may exist, randomly choosing compression target has sufficient flexibility and variability, thereby enabling language model to  compress diverse range of contextual information. At inference time, we currently adopt the same random strategy as training time.
>
> **Q**: "Can this method play well with Flash Attention to improve inference even further? If no, how your method compare with flash attention?"
> **A**: The original Flash Attention implementation requires the attention mask matrix to be lower triangular, meaning that it is not directly compatible with our method due to the sparse attention mask. To combine our method with efficient attention implementation, we need to use more advanced kernels that support sparsity patterns in attention masks. One such implementation is sparse flash attention[1]. Xformers[2]' implementation of memory-efficient attention also supports an arbitrary attention mask, one can directly use it to replace the default attention implementation in HuggingFace.
> References:
> [1] Faster Causal Attention Over Large Sequences Through Sparse Flash Attention.
> [2] xFormers: A modular and hackable Transformer modeling library.
>
> **Q**: "No mention of open-sourcing implementation"
> **A**: We have uploaded the source code in the supplementary material, including the main Python files and Bash scripts. We will open-source it after the paper is accepted.

---

### Meta-Review · Area_Chair_GFDd · 2023-09-18

**Recommendation:** 3

**Metareview:**

This paper presents a novel method aimed at compressing the memory consumption of Key-Value (KV) cache and reducing the computation time of attention in decoder Transformer models. The approach leverages the use of two sentinel tokens, <CL> and <CR>, which encapsulate spans of tokens to compress. By subsequently fine-tuning the model, it compresses the information between these two tokens into one, enabling memory savings.

In general, the reviewers were in consensus about the novelty of the and were generally positive on the empirical results compared to prior work.

There were some methodological concerns related to how the improvement of the approach compared to existing work in practical terms. For instance, while KV Compression was compared against Local Attention in terms of compression ratio and performance metrics (e.g., perplexity), they were not compared in terms of throughput. Instead, throughput is provided only for the proposed method.

Additionally, one reviewer felt that the datasets and models evaluated in the initial submission were limited. The authors provided additional results that satisfied the reviewer.

---

### Decision · Program_Chairs · 2023-10-07

**Decision:**

Accept-Main

**Comment:**

This paper presents a novel method aimed at compressing the memory consumption of Key-Value (KV) cache and reducing the computation time of attention in decoder Transformer models. The approach leverages the use of two sentinel tokens, <CL> and <CR>, which encapsulate spans of tokens to compress. By subsequently fine-tuning the model, it compresses the information between these two tokens into one, enabling memory savings.

In general, the reviewers were in consensus about the novelty of the and were generally positive on the empirical results compared to prior work.

There were some methodological concerns related to how the improvement of the approach compared to existing work in practical terms. For instance, while KV Compression was compared against Local Attention in terms of compression ratio and performance metrics (e.g., perplexity), they were not compared in terms of throughput. Instead, throughput is provided only for the proposed method.

Additionally, one reviewer felt that the datasets and models evaluated in the initial submission were limited. The authors provided additional results that satisfied the reviewer.